# The Interaction Between Self-Efficacy, Fear of Failure, and Entrepreneurial Passion: Evidence from Business Students in Emerging Economies

**DOI:** 10.3390/bs15070951

**Published:** 2025-07-14

**Authors:** Elizabeth Emperatriz García-Salirrosas, Dany Yudet Millones-Liza, Rafael Fernando Rondon-Eusebio, Jorge Alberto Esponda-Pérez, Eulalia Elizabeth Salas-Tenesaca, Reinaldo Armas-Herrera, María Fernanda Zumba-Zúñiga

**Affiliations:** 1Faculty of Management Science, Universidad Autónoma del Perú, Lima 15842, Peru; 2Universidad Tecnológica del Perú, Lima 15487, Peru; c29322@utp.edu.pe; 3Department of Humanities, Universidad Privada del Norte, Lima 15314, Peru; rafael.rondon@upn.edu.pe; 4Faculty of Nutrition and Food Sciences, Universidad de Ciencias y Artes de Chiapas, Tuxtla Gutiérrez 29000, Mexico; jorge.esponda@unicach.mx; 5Departamento de Ciencias Empresariales, Facultad de Ciencias Económicas y Empresariales, Universidad Técnica Particular de Loja, Loja 1101608, Ecuador; eesalas@utpl.edu.ec (E.E.S.-T.); ahreinaldo@utpl.edu.ec (R.A.-H.); mfzumba@utpl.edu.ec (M.F.Z.-Z.)

**Keywords:** risk perception, achievement motivation, cognitive mediators, business education, academic entrepreneurship, developing markets

## Abstract

This study examines the relationship between fear of failure, entrepreneurial passion, and entrepreneurial self-efficacy among university business students from four emerging Latin American economies: Mexico, Colombia, Peru, and Ecuador. This research addresses the need to understand the psychological factors that influence entrepreneurship, particularly in developing economies. Using non-probability convenience sampling, surveys were distributed between June 2021 and August 2022, and 961 valid responses were obtained. Scales from renowned researchers were used, which were translated and semantically validated in Spanish to measure the three variables. The data were analyzed using structural equation modeling with PLS. The overall results reveal that fear of failure negatively affects entrepreneurial passion, while entrepreneurial passion has a positive and significant relationship with entrepreneurial self-efficacy. Furthermore, entrepreneurial passion has a positive and significant relationship with fear of failure and entrepreneurial self-efficacy, with substantial differences between countries. Gender differences were also identified: in women, all relationships in the model were significant, while in men, the direct effect of fear of failure on entrepreneurial self-efficacy did not reach statistical significance. These findings contribute to the entrepreneurship literature by demonstrating that the relationship between these psychological variables varies by context (country) and gender, contradicting the notion of a universal effect.

## 1. Introduction

Entrepreneurship is a key driver of economic growth and sustainable development, particularly in emerging economies ([18]; [31]). Conceptualized as any effort to develop new businesses or enterprises, entrepreneurship involves a complex and dynamic activity that requires individuals to engage in cognitive processes that consider future outcomes ([29]). Several studies have focused on identifying the factors that determine entrepreneurial success or failure. Over the past decade, research attention has shifted toward understanding the sociological identities of entrepreneurs, including entrepreneurial passion and fear of failure. The best entrepreneurs learn from their experiences, persist, and try again ([38]). In addition to these studies, recent contributions have emphasized the significance of passion and positive emotions in individuals who aspire to start a business. These qualities are often more crucial than technical knowledge and skills for engaging in committed and dedicated commercial activities ([59]; [100]). Some authors have conceptualized this trait as “entrepreneurial passion”, which involves intense, positive, and consciously accessible feelings experienced when engaging in entrepreneurial activities, primarily associated with role identity ([32]). In contrast, “fear of failure” is a significant barrier to entrepreneurship as it reduces perceptions of potential success in the intended activity. Likewise, a study suggests that in economies with higher levels of economic freedom, the negative impact of fear of failure is mitigated by more alternative opportunities for entrepreneurs, thus reducing the fear of failure ([27]).

People’s willingness to undertake an activity is influenced by several factors, including the fear of failure, which in turn leads to lower probabilities of success in their ventures ([54]; [66]), because it constitutes a psychological construct that explaind the interaction between the cognitive, emotional, and motivational attributes of entrepreneurs in the context of achievement ([92]). This psychological barrier significantly diminishes entrepreneurial motivation and endangers entrepreneurial self-efficacy ([14]; [81]). Recent research acknowledges that fear of failure is an inevitable sensation within entrepreneurial actions and represents a high psychological barrier that limits entrepreneurial self-efficacy ([5]; [81]; [85]).

However, fear of failure can be mitigated by a positive attitude towards starting business activities, even when resources are limited ([46]; [52]; [93]). A positive attitude, proactive personality, task competence, and belief in one’s capabilities to succeed in entrepreneurship are crucial to entrepreneurial self-efficacy, which is associated with an individual’s intention to become an entrepreneur ([7]; [12]; [24]; [60]; [96]).

Entrepreneurial self-efficacy is a topic of significant interest that reveals its relationship with entrepreneurial passion and fear of failure, factors that are more representative in emerging and developing economies such as Mexico, Peru, Colombia, and Ecuador, which were part of this study and together constitute an important economic group called the Pacific Alliance ([30]; [88]). This applies particularly to Ecuador, which is seeking to join the Alliance, given that its formalization processes are almost complete. The countries that belong to the Pacific Alliance, except Chile, have given way to the consolidation of a crucial economic block that represents 35% of the total GDP of Latin America and 2.7% of the world GDP, generating 50% of Latin American trade and 49% of the region’s exports, thus becoming an important global economy that boosts the entrepreneurial sector in Latin America ([37]). However, we must point out that these nations represent low-income economies of less than $25,000 a year, with an average GDP growth rate of 1.53% in 2017 Global Entrepreneurship Monitor ([38]). Hence, the primary motivation for entrepreneurship is the scarcity of employment, which contributes to improving employment rates. In 2023, Latin American countries exhibited the highest rates of early entrepreneurial activity, with an average TEA of 25.76%, and Ecuador led the region with the highest TEA. However, these economies are characterized by developing entrepreneurial ecosystems that are not yet consolidated or mature ([58]), thus highlighting the need to promote entrepreneurial education programs to provide students with essential knowledge and skills for entrepreneurship, as well as strengthen psychological factors to increase their ability to participate in entrepreneurial activities ([26]).

According to [28] ([28]), 90% of startups do not achieve sustainability. This explains the negative feelings of fear that arise in entrepreneurs, which undermine entrepreneurial self-efficacy. Emerging within this dynamic, entrepreneurial passion is a fundamental element that softens the negative impact of fear on self-efficacy, thus facilitating the entrepreneur’s perseverance ([89]).

Specifically, business students have a special advantage in that their academic training allows them to develop skills that influence their attitudes towards entrepreneurship, which also improves their entrepreneurial self-efficacy, as it has been shown that problem-based learning courses and the development of topics related to business planning drive entrepreneurial intentions by improving self-efficacy in students ([13]; [79]; [22]). Even exposure to successful entrepreneurial models during academic training significantly improves confidence; in contrast, fear of failure emerges as an effect of academic procrastination and avoidance behaviors ([16]; [63]). A critical antecedent that explains entrepreneurial passion in business students is that it is associated with the recognition of opportunities to start new businesses, added to previous knowledge and experience, so that educational institutions should promote entrepreneurial education programs that, beyond teaching entrepreneurial skills, can cultivate entrepreneurial passion ([59]; [93]).

Similarly, previous studies report that when a student has low self-efficacy, this could lead to a greater fear of failure, since he or she doubts his or her ability to succeed; in this context, fear of failure generates an adverse effect on entrepreneurial self-efficacy and, in addition, it can be attenuated by entrepreneurial passion ([51]; [70]). Therefore, decreasing negative emotions could correct the attribution to failure and increase self-efficacy; to that effect, fostering practical experience in students to operate in real scenarios would be an assertive action ([53]).

Although there is academic interest in examining the relationship between these psychological factors and self-efficacy, little is known about the implications of the current research in the context of the Pacific Alliance countries. This aspect opens up a wide range of possibilities for exploring this fact, given that the social, economic, and cultural environments of entrepreneurs in these economies may vary.

For this reason, this study aims to uncover the dynamics between the study variables to reinforce in educational centers the need to pay more attention to entrepreneurial education in curricular designs, the consolidation of incubation and mentoring programs, and other mechanisms that strengthen entrepreneurial passion, as it is known that this can neutralize the link between fear of failure and entrepreneurial self-efficacy. In this context, the question arises: Can entrepreneurial passion mediate the relationship between fear of failure and entrepreneurial self-efficacy? Does fear of failure hurt entrepreneurial self-efficacy? Does entrepreneurial passion mediate the relationship between fear of failure and entrepreneurial self-efficacy? Does entrepreneurial passion mediate the relationship between fear of failure and entrepreneurial self-efficacy? To answer this question, this research aims to develop a predictive model that examines the influence of fear of failure and entrepreneurial passion on the entrepreneurial self-efficacy of business students in four countries: Mexico, Colombia, Peru, and Ecuador.

## 2. Theoretical Background

### 2.1. Research Variables

#### 2.1.1. Fear of Failure

It is a construct that manifests itself as a psychological obstacle that people manifest before initiating or developing an action. In this process, individuals pay greater attention to their negative experiences and failed actions, which decreases their hope and optimism ([54]; [66]). Thus, it has been found that many entrepreneurs opt to become employees instead of continuing their entrepreneurial ventures, as they prefer a guaranteed salary instead of taking a risk ([56]). According to previous studies, fear of failure can shape business results, influencing decision-making and the willingness to take risks. In that sense, to mitigate the feeling of fear of failure, self-confidence has been considered an important attribute, since it has been shown that positive attitudes increase the probability of success among entrepreneurs ([74]; [84]).

Thus, studies have shown that one of the fundamental characteristics of an entrepreneur is self-confidence and the ability to make decisions without fear of failure ([74]). Therefore, every entrepreneur must work on these two fundamental aspects to face the challenges of the commercial world and take advantage of opportunities with a resilient and proactive mindset ([95]).

A theory that deepens the fear of failure is centered on achievement motivation theory, which states that individuals simultaneously evaluate the expectation of success and the fear of embarrassment of failure before acting. According to this theory, individuals who maintain a high degree of fear of failure usually avoid tasks with a risk of negative outcomes. In this sense, achievement motivation theory highlights an avoidance component that reduces entrepreneurial intentions ([77]). Meanwhile, the theory of protective motivation establishes that fear of failure is a factor that motivates individuals to adopt behaviors as a protective alternative to perceived threats ([47]).

[21] ([21]) Focuses on the fear of failure using the Performance Failure Appraisal Inventory (PFAI), which presents five factors: (a) fear of upsetting important people, referring to the fear of upsetting or disappointing another person whom one values; (b) fear of self-esteem, here emerges the belief that failure represents incapacity; (c) fear of shame, referring to the fear that arises when a person exposes him/herself publicly before others who might judge him/her; (d) fear of an unknown future, i.e., anxiety emerges due to the fear that failure may alter some projected plans; and (e) fear of significant people losing interest, in this dimension, the fear that other people may lose interest due to failure is denoted. Thus, this study aims to assess the threat appraisals associated with fear of failure, specifically in the context of performance appraisal and achievement situations.

#### 2.1.2. Entrepreneurial Passion

It is the positive attitude towards entrepreneurial activities that an individual develops when immersed in the creation and management of a business. Moreover, it is a crucial element that identifies an entrepreneur, as it is related to a strong positive feeling for executing entrepreneurial activities with energy, even when resources are limited ([46]; [52]; [93]). It is also defined as an intense and positive talent and feeling that enables productive and successful ventures that contribute to society ([46]; [39]; [73]). Entrepreneurial passion is an innovative and passionate characteristic of entrepreneurs that makes them successful in seeking new opportunities where business flows and meets different market demands ([101]). Accordingly to the afore metioned, [87] ([87]) state that entrepreneurial passion is the force that helps entrepreneurs identify market signals to take advantage of opportunities, acting as a catalyst that strengthens entrepreneurial motivation and intention, which, combined with curiosity, has a favorable effect on learning and achieving long-term goals ([15]; [102]).

A review of the theories that support entrepreneurial passion identified the Dual Passion model, which posits that not all passion is the same in individuals, as it depends on how entrepreneurial activities have been integrated into their identity. According to this theory, there are two types of passion: harmonious and obsessive. The former is associated with favorable results, such as persistence and success, whereas the latter can reduce passion when adverse situations arise ([36]; [48]). Another theory that supports entrepreneurial passion is the Theory of Planned Behavior, which states that attitudes, subjective norms, and perceived behavioral control influence an individual’s intentions, such that the appropriate action of entrepreneurial support and self-efficacy provides exceptional support to entrepreneurial passion ([103]; [68]).

#### 2.1.3. Entrepreneurial Self-Efficacy

It is part of an entrepreneurial action that originated from a proactive personality. It is an essential socio-cognitive process in the intentions and behavior of an entrepreneur ([10]; [17]; [60]; [96]). Additionally, it is considered the judgment a person has regarding his or her ability to perform some task in his or her domain, as well as the belief in his or her ability to achieve success in a venture ([12]; [24]). It is also defined as the psychological trait associated with an individual’s intention to be an entrepreneur ([7]) and their level of confidence in their entrepreneurial capacity ([86]). In this context, [71] ([71]) established that self-efficacy has become a central component of social cognitive theory that promotes the willingness to take on challenges and meet goals, so that its impact is significant and positive in business ([8], [9], [11]; [82]; [94]). Moreover, previous studies have shown that to acquire entrepreneurial self-efficacy, it is necessary to identify the barriers and facilitators that affect the development of self-confidence, face entrepreneurial challenges, and increase the probability of success ([20]).

The theory of entrepreneurial self-efficacy posits that it serves as a catalyst that translates an entrepreneur’s knowledge into action. In this dynamic, the confidence of an individual in their ability to perform ([12]; [76]) Furthermore, it is based on socio-cognitive theory and outcome expectancy; in the former, learning arises as a result of the interaction between personal factors, behavioral patterns, and environmental influences ([82]); and in the latter, a positive expectancy motivates individuals to adopt behaviors projected to obtain desired results ([94]).

Considering that each of the concepts involving the study variables addresses a crucial issue within the business environment, where young people symbolize the key force for economic and social development, it becomes essential to promote entrepreneurship. For this purpose, it is necessary to know which factors could represent a barrier or obstacle for young people who want to become entrepreneurs. To create a successful and sustainable entrepreneurial culture in countries with emerging economies, where political instability, inequalities, and other financial factors hinder young people’s capacity to take on challenges, it is crucial to explore the psychological factors that may contribute to the reduction or failure of entrepreneurship. Two factors have been considered for this purpose: fear of failure and self-efficacy. The following section provides a more in-depth explanation of the associations between these factors.

### 2.2. Conceptual Model and Research Hypotheses

#### 2.2.1. Association Between Fear of Failure and Entrepreneurial Self-Efficacy

Fear of failure reduces the probability of achieving success and good outcomes. Thus, it is considered a negative feeling that dulls entrepreneurial motivation and calls into question the entrepreneurial self-efficacy ([36]). This represents a psychological barrier that limits high expectations for the fulfillment of set goals ([90]). Therefore, fear of failure has been considered an inevitable feeling in entrepreneurial actions that arises spontaneously and/or as a consequence of internal and/or external pressure, weakening motivation, and eroding entrepreneurial self-efficacy. This means that the environment creates fear of failure, thus affecting entrepreneurial self-efficacy ([14]; [81]).

Some studies have established that a high perception of entrepreneurial self-efficacy can mitigate the adverse effect of fear of failure, which represents an individual’s security regarding their ability to undertake entrepreneurial activities ([81]; [85]). Likewise, other studies suggest that fear of failure hinders the promotion of entrepreneurship. Therefore, strengthening entrepreneurial self-efficacy would lead to counteracting the barriers of fear of failure ([4]). Fear of failure constitutes an obstacle to entrepreneurial possibilities because it makes individuals adopt actions and behaviors that limit their initiative, creativity, and knowledge, negatively affecting their entrepreneurial self-efficacy ([26]). And it has even been identified that university students, even when they receive constant training, present a feeling of fear, the same that attenuates entrepreneurial self-efficacy; thus, the fear of failure becomes a constraint for students to enhance their entrepreneurial self-efficacy ([55]; [70]). Based on the arguments presented, the following hypothesis is proposed:

**H1.** 
*Fear of failure has a significant negative effect on entrepreneurial self-efficacy.*


#### 2.2.2. Association Between Fear of Failure and Entrepreneurial Passion

Passion awakens creativity, stimulates participation, and optimizes entrepreneurial behavior ([67]). In contrast, the fear of failure could be a determinant of entrepreneurial passion at risk, and due to the fear of failure, it tends to reduce entrepreneurial passion, since the negative feeling allows an individual to perceive the low probability of success in the face of an opportunity ([6]; [61]). In addition, other studies have reported that when an individual feels fear, their passion and/or motivation decreases ([64]). This means that when perceiving a threatening or uncertain situation, attention is shifted to focus on possible losses, and negative emotions are activated, affecting the energy available to experience enjoyment, curiosity, and passion for an activity.

Thus, any support system is appropriate to mitigate the negative impact of fear of failure on entrepreneurial passion because, according to previous studies, to reduce the adverse effects of fear of failure, it is necessary to create resilient enabling environments that preserve entrepreneurial passion ([64]; [91]). This same dynamic is also associated with university students, as it has been shown that fear of failure negatively impacts feelings related to entrepreneurship; this means that fear of failure acts as a psychological barrier that hinders entrepreneurial behavior ([36]). These considerations lead to the following hypothesis:

**H2.** 
*Fear of failure has a significant negative effect on entrepreneurial passion.*


#### 2.2.3. Association Between Entrepreneurial Passion and Entrepreneurial Self-Efficacy

Some antecedents suggest that entrepreneurial passion has a positive effect on entrepreneurial self-efficacy. That is, the higher the entrepreneurial passion, the higher the entrepreneurial self-efficacy ([81]). In addition, other studies have established that entrepreneurial passion serves as a measure of an entrepreneur’s persistence and resilience. Therefore, passion generates greater viability in reinforcing entrepreneurial self-efficacy when facing challenges, becoming a key element in each business activity ([98]; [97]).

Studies have also identified that entrepreneurial passion plays a vital role in strengthening entrepreneurial self-efficacy and increasing the probability of success among entrepreneurs ([97]). Thus, the credibility that an entrepreneur has in themselves gives openness to a special dynamic that arouses passion, expressed in the commitment to face challenges, leading to significant entrepreneurial growth ([25]). Therefore, it is clear that entrepreneurial passion enhances entrepreneurial self-efficacy, as it drives the motivation to innovate and adapt to market needs. This is a reality for university business students, as it is known that when they receive an entrepreneurial education, they tend to develop a greater passion for entrepreneurial activities, which strengthens their skills to the point of having the ability to perform tasks and other functions related to entrepreneurship ([6]; [44]). The paragraphs above establish the following hypothesis:

**H3.** 
*Entrepreneurial passion has a significant positive effect on entrepreneurial self-efficacy.*


#### 2.2.4. Role of Entrepreneurial Self-Efficacy in the Association of Fear of Failure and Entrepreneurial Passion

In the dynamic world of business, entrepreneurial passion emerges as a key driver propelling entrepreneurs to overcome obstacles ([36]). However, what happens when fear of failure appears in entrepreneurial intentions? Studies have shown that entrepreneurial self-efficacy is a crucial element in mitigating fear of failure ([3]; [17]). Thus, self-efficacy enables entrepreneurial intention to consolidate into real and sustainable actions. At the same time, it awakens the potential for entrepreneurial passion and closes the gap between an entrepreneurial project and its execution, reducing the effect of fear of failure ([78]). That is, an entrepreneur who is confident in their capacity for self-efficacy can mitigate fear and insecurity, transforming uncertainty into an opportunity for learning and growth ([43]).

Although research suggests that fear of failure can limit entrepreneurial passion, some studies indicate that self-efficacy can play a mitigating role in overcoming the barrier imposed by fear, thereby generating a positive mindset ([3]; [17]). In summary, entrepreneurial self-efficacy is a crucial component that empowers entrepreneurs with strength, motivation, commitment, security, and confidence, thereby enhancing their entrepreneurial passion. In other words, by believing in their abilities to face challenges and overcome obstacles, entrepreneurs feel empowered to pursue their entrepreneurial goals effectively, generating entrepreneurial passion and reducing their fear of failure. In addition, previous studies have reported that fear of failure hurts entrepreneurial self-efficacy; that is, students who fear failure tend to have less confidence in their entrepreneurial skills and therefore their self-efficacy is low; In contrast, when a student is passionate about entrepreneurship, he/she consolidates his/her skills better ([61]; [69]). In this context, the following hypothesis is proposed:

**H4.** 
*Entrepreneurial passion mediates the relationship between fear of failure and entrepreneurial self-efficacy.*


In general, studies that have evaluated self-efficacy support its effectiveness in addressing personal and professional challenges. The greater the self-efficacy, the better prepared people are to face challenges and achieve success in their ventures ([49]; [65]; [99]). In the same line of research, previous studies have identified that entrepreneurial passion is a factor that offers a notable boost to achieve the emotional experience that facilitates business opportunities and favors the entrepreneurial environment ([2]; [36]). A favorable environment allows productive and creative business activities. However, when conditions are uncertain, fear of failure emerges, which inhibits entrepreneurial initiative and puts business self-efficacy at risk ([92]). Based on this, the leading role of these variables in entrepreneurship and small businesses, on which the advancement of society, job creation, and the economic growth of a country largely depend, is highlighted ([72]).

The analyzed variables provide a clear overview of the factors that are part of the business ecosystem. Despite the great relevance of this topic, only five studies have been identified. The first one, conducted in China, of a longitudinal nature, has a greater emphasis on female entrepreneurial passion ([35]). The second study was carried out in China, which focuses on obsessive passion as a promoter of business performance ([92]). The third study conducted in India focused on understanding the role of business education and fear of failure in entrepreneurship ([90]). In the fourth study, the authors investigated the factors that facilitate entrepreneurial motivation in 23 Organisation for Economic Co-operation and Development (OECD) countries ([36]).

Finally, a fifth study carried out in China analyzes the behavior of entrepreneurs in the face of a crisis ([50]). The countries where these studies were conducted have developed economies. Thus, the results cannot be extrapolated to a developing economy. From the above, we notice a research gap in the educational context that urgently needs exploration. In this context, it is assumed that entrepreneurship programs can be directed and supervised by experienced teachers in the field. This orientation is essential given the need for knowledge about entrepreneurship strategies, especially in countries with emerging economies, where resources are limited and conditions are unstable ([75]; [80]).

Based on the information presented in this chapter, a theoretical model is proposed, as illustrated in Figure 1. Four hypotheses (H1, H2, H3, and H4) are proposed to validate the model, establishing direct and indirect relationships between three variables in entrepreneurial behavior: Fear of Failure (FF). It plays the role of an independent variable (exogenous), negatively related to Entrepreneurial Self-Efficacy (ESE) (H1) and Entrepreneurial Passion (EP) (H2), indicating that a lower FF is associated with higher ESE and EP. Entrepreneurial Passion (EP). It plays several roles in the proposed model, as follows. It is a dependent variable in a negative relationship with FF (H2) and an independent variable in a positive relationship with SES (H3). It also mediates FF, which affects SES (H4). Entrepreneurial Self-Efficacy (ESE). It acts as a dependent variable (endogenous), receiving impacts from both the negative relationship with FF (H1) and the positive relationship with EP (H3). The theoretical model aims to demonstrate how Entrepreneurial Passion can serve as a protective factor that mitigates the potentially harmful effects of Fear of Failure on Entrepreneurial Self-Efficacy, underscoring the importance of utilizing mechanisms that foster passion in the training and development processes of entrepreneurs.

## 3. Research Methodology

### 3.1. Sample and Procedure

This study was approved by the Ethics Committee of the Postgraduate Studies Department of Universidad Peruana Unión (2021-CE-EPG-00011). Informed consent was obtained from the participants before administering the survey. Non-probabilistic convenience sampling was used. This type of non-probabilistic convenience sampling is suitable for obtaining data quickly and cheaply. In addition, this method is practical when participants are accessible and willing to collaborate, allowing researchers to move forward with their studies without significant logistical restrictions ([62]). The survey was conducted via a virtual link that redirected participants to a questionnaire hosted on Google Forms. The survey was distributed via email and WhatsApp to all undergraduate and postgraduate business students. The respondents’ features were diverse and partly due to their use of social networks and the Internet ([83]). To evaluate consumer behavior, some authors recommend a sample size greater than 100 subjects or five times the number of items ([34]; [40]). The questionnaire used in the present study consisted of 31 essential items and 4 sociodemographic items (Gender, Country, Age, and Academic Level), totaling 35 items. Therefore, a sample size of 350 was considered adequate. Ultimately, 961 subjects were included, which was sufficient for the present investigation. Information was collected from Latin American business students pursuing higher education. From Colombia (219 responses, 22.8%), Ecuador (230 responses, 23.9%), Mexico (236 responses, 24.6%), and Peru (276 responses, 28.7%). Data were collected from June 2021 to August 2022. Table 1 presents the sociodemographic data regarding gender, country, age, and academic level.

### 3.2. Instrument

Three scales validated in previous research were used for data collection. In this sense, fear of failure was evaluated using a measurement model based on five dimensions: (a) fear of upsetting important people; (b) fear of self-esteem; (c) fear of shame; (d) fear of an unknown future; and (e) fear of significant people losing interest ([21]). Similarly, to assess entrepreneurial self-efficacy, an adaptation was made based on two studies that evaluated this construct ([23]; [57]). At the same time, to assess entrepreneurial passion, a scale constructed to evaluate this construct by measuring entrepreneurs’ intense positive feelings was adapted ([19]). Since the original scales were in English, a rigorous translation process into Spanish was implemented for this research, followed by a semantic validation carried out by a team of six researchers to verify the comprehension of the items in the four countries. Subsequently, a pilot survey was administered to 40 participants per country, and a reliability analysis was carried out using Cronbach’s alpha coefficient ([1]). As the reliability results of the questionnaire were adequate (above 0.7), validity and revalidation analyses demonstrated the appropriate performance of the translated scale. Likert-type scales with different ranges were used to evaluate the study variables. A 5-point scale was used to measure fear of failure, where 1 represents “Strongly Disagree” and 5 means “Strongly Agree”. On the other hand, for the evaluation of self-efficacy and entrepreneurial passion, a 7-point scale was used, where 1 corresponded to “Strongly Disagree” and 7 to “Strongly Agree”. For details of the items, see Table A1 in Appendix A.

To reduce the influence of social desirability bias, we ensured that the participants’ responses were completely anonymous during the data collection process, and we used neutral wording for all items with great care. These methodological safeguards were implemented to promote honest responses from participants, instead of socially desirable self-presentation. The research design fostered an environment that promoted genuine disclosure and improved the validity of the data gathered by removing participant identification and employing non-inductive questioning.

### 3.3. Statistical Procedures

Partial Least Squares (PLS), a variation-based structural equation modeling (SEM) method, was used to examine the data. As an SEM technique, PLS enables the calculation of the links between theoretical constructs and allows for the simultaneous assessment of theoretical construct validity and reliability. Furthermore, PLS is advantageous because this technique employs manifest variables, although not directly observable, to represent a construct ([42]). Because of its first-order construct shape, the second-order construct (Fear of Failure) was generated formatively as a composite.

## 4. Results

### 4.1. Measurement Model Assessment

To perform the PLS analysis, it was necessary to verify the reliability and validity (convergent and discriminant) of all the scales used in the proposed model. In a preliminary study, several items that did not meet the required reliability indicators were identified and removed. Specifically, two items corresponding to the dimension ‘Fear of shame’ (FF1 and FF2), one item from the dimension ‘Fear of an uncertain future’ (FF15), and one item from the dimension ‘Fear of important people losing interest’ (FF18) were excluded. In total, four items were eliminated from the original fear of failure scale, which consisted of 25 items, leaving 21 items. Subsequent analyses were performed considering only retained items.

#### 4.1.1. Convergent Validity and Internal Consistency

Table 2 presents the results obtained from the 961 valid responses, incorporating both the reliability indicators for each construct and the descriptive analyses of the items (mean, standard deviation (SD), skewness (SK), and kurtosis (K)). The analysis of the factor loadings shows that all the model items obtained values higher than 0.7, evidencing a high reliability of the indicators concerning their corresponding constructs. Likewise, all the constructs met the methodological criteria established in the specialized literature ([41]): average variance extracted (AVE) greater than 0.5, and both the composite reliability (CR) and Cronbach’s alpha with values greater than 0.70 ([41]). These findings confirm that the indicators reached satisfactory values for the first-order constructs, thus validating the internal consistency and reliability of the items included in the proposed model.

#### 4.1.2. Discriminant Validity Assessment

Discriminant validity was tested following the Fornell-Larcker criteria ([33]), using the square root of the AVE of each latent variable for the first-order constructs. Since the AVE square roots (quadratic values in bold) were greater than the correlation coefficient between the latent variables, we could determine the existence of discriminant validity. These findings attest to the reliability and discriminant validity of these scales across constructs ([42]). The variables met the estimate, and each one of them represented a unique construct. Table 3 displays the discriminant validity of the first-order constructs. Fear of Failure is a consistent construct, as evidenced by the formative component of the model for the general sample (961 valid responses), where first-order constructs had a statistically significant impact on the second-order construct “Fear of Failure”.

To complement the discriminant validity analysis of the model, the heterotrait-monotrait ratio (HTMT) criterion was applied. This more rigorous method provides a robust assessment of discriminant validity by evaluating the distinctiveness of the constructs associated with each factor ([45]). This technique is particularly appropriate for assessing hierarchical component models that are organized by levels. According to the standards established in the specialized literature, the HTMT coefficients must be below the strict threshold of 0.850 to confirm discriminant validity. The results obtained through this supplementary analysis are presented in detail in Table 4, where all values satisfactorily meet this criterion, thus confirming the discriminant validity of the proposed model.

### 4.2. Hypothesis Testing Results

The results confirm all hypotheses proposed in the theoretical model. The strongest relationship is the positive influence of entrepreneurial passion on entrepreneurial self-efficacy (H3: β = 0.620; *p* < 0.001), followed by the adverse effect of fear of failure on entrepreneurial passion (H2: β = −0.112; *p* < 0.001). The mediating effect (H4: β = −0.069; *p* < 0.001) reveals an essential indirect mechanism through which fear of failure decreases entrepreneurial self-efficacy by first reducing it. Although statistically significant, the direct effect of fear of failure on entrepreneurial self-efficacy (H1: β = −0.058; *p* = 0.037) is the weakest of the relationships analyzed (See Table 5).

#### Structural Model Assessment and Effect Sizes

Figure 2 graphically shows the theoretical model evaluated using structural equations, illustrating the connections between the variables. The model presents Fear of Failure as a second-order construct composed of five dimensions: Fear of Shame, Fear of Loss of Self-esteem, Fear of an Uncertain Future, Fear of Important People Losing Interest, and Fear of Disappointing Important People, showing a complex hierarchical structure.

Each fear dimension is represented by its respective indicators (FF3–FF25) and factor loadings, which range from 0.739 to 0.923, demonstrating adequate convergent validity. The model shows a structural relationship in which fear of failure is directly related to entrepreneurial passion (with a coefficient of −0.058, *p* = 0.037) and indirectly through entrepreneurial passion.

Entrepreneurial Passion, with an R^2^ of 0.012 shown in the blue circle, is measured by four indicators (EP1–EP4) with very high factor loadings (0.904–0.948). This variable has a significant influence on Entrepreneurial Self-Efficacy, with a coefficient of 0.620 (*p* = 0.000). The effect size (f^2^) for this relationship is 0.629, indicating a significant effect according to Cohen’s criteria (f^2^ ≥ 0.35 for substantial effects). Entrepreneurial Self-Efficacy, with an R^2^ of 0.396, as noted in the blue circle, is measured by six indicators (ESE1–ESE6) with factor loadings ranging from 0.813 to 0.896. The effect size analysis reveals that Fear of Failure has a negligible effect on both Entrepreneurial Passion (f^2^ = 0.013) and Entrepreneurial Self-Efficacy (f^2^ = 0.005), suggesting that while statistically significant, the practical significance of fear of failure as a predictor is limited compared to the substantial influence of entrepreneurial passion on self-efficacy.

Additionally, the predictive significance analysis using the Stone-Geisser indicator (Q^2^) yielded positive values of 0.010 for Entrepreneurial Passion and 0.014 for Entrepreneurial Self-Efficacy, confirming the model’s specific predictive ability. The results suggest that Fear of Failure has a direct adverse effect on Entrepreneurial Passion, which in turn positively influences Entrepreneurial Self-Efficacy. This finding highlights the role of Entrepreneurial Passion as a mediator in the relationship between Fear of Failure and entrepreneurial self-efficacy among business students from emerging economies.

### 4.3. Multi-Group Analysis by Gender

A gender-differentiated analysis was also performed to examine possible variations in the model’s relationships. The results revealed a non-table difference between both groups: while in the female sample all hypotheses were confirmed, in the male sample it was observed that the negative influence of fear of failure on entrepreneurial self-efficacy (b = −0.051; *p* = 0.269) did not reach statistical significance, leading to the rejection of hypothesis H1 only for this group. Therefore, the results support hypotheses H1, H2, H3, and H4 in the female sample, while in the male sample, only hypotheses H2, H3, and H4 were confirmed. Table 6 presents all these comparative findings by gender.

### 4.4. Cross-Country Analysis

The analysis by country in the Colombian sample revealed that fear of failure did not significantly influence entrepreneurial self-efficence (β = −0.079; *p* = 0.064) or entrepreneurial passion (β = 0.036; *p* = 0.595). While entrepreneurial passion has a significant influence on entrepreneurial self-efficacy (β = 0.740; *p* = 0.000), this is the strongest relationship in the model for Colombia. Furthermore, entrepreneurial passion does not mediate the relationship between fear of failure and entrepreneurial self-efficacy (β = 0.027; *p* = 0.596). These results do not support H1, H2, and H4; only H3 was supported.

The results for the Ecuadorian sample are similar, where fear of failure did not influence entrepreneurial self-efficacy (β = −0.043; *p* = 0.523). The effect of fear of failure on entrepreneurial passion is insignificant (β = −0.127; *p* = 0.063). Although Entrepreneurial passion significantly influences entrepreneurial self-efficacy (β = 0.450; *p* = 0.000), the mediating effect is insignificant (β = −0.057; *p* = 0.073). These results support hypothesis H3 and reject H1, H2, and H4.

In Mexico, there is no significant direct effect of fear of failure on entrepreneurial self-efficacy (β = −0.086; *p* = 0.155); however, fear of failure has a substantial adverse impact on entrepreneurial passion (β = −0.260; *p* = 0.000), which is the most considerable effect among all countries. Similarly, entrepreneurial passion has a significant influence on entrepreneurial self-efficacy (β = 0.570; *p* = 0.000). The mediating effect is significant (β = −0.148; *p* = 0.002), and is the strongest among the four countries. These results support hypotheses H2, H3, and H4; only H1 is rejected.

The results for Peru are similar to those for Mexico: there is no significant direct effect of fear of failure on entrepreneurial self-efficacy (β = −0.054, *p* = 0.401). However, fear of failure has a significant negative effect on entrepreneurial passion (β = −0.202; *p* = 0.000). Entrepreneurial passion also significantly influences entrepreneurial self-efficacy (β = 0.437; *p* = 0.000), and the mediating effect is significant (β = −0.088; *p* = 0.000). These results support hypotheses H2, H3, and H4, with only H4 being rejected.

A comparison between countries shows the following results: Hypothesis H3 is the only hypothesis consistently confirmed in all countries. H1 is not confirmed in any country. Mexico shows the most potent effects on the negative relationship with fear of failure. Colombia presents a distinctive pattern, where H2 has a positive (though not significant) coefficient. Mediation (H4) is substantial in Mexico and Peru, marginally significant in Ecuador, and insignificant in Colombia. These results suggest essential cultural or contextual differences in how these psychological relationships operate among business students in the different Latin American countries studied. The results are presented in Table 7.

## 5. Discussion

This research aims to develop a predictive model to investigate the impact of fear of failure and entrepreneurial passion on the entrepreneurial self-efficacy of business students in four countries: Mexico, Colombia, Peru, and Ecuador. The results reveal a complex dynamic that varies according to country context and gender, providing valuable insights into understanding entrepreneurial behavior in emerging economies. This research aims to develop a predictive model to investigate the impact of fear of failure and entrepreneurial passion on the entrepreneurial self-efficacy of business students in four countries: Mexico, Colombia, Peru, and Ecuador. The results reveal a complex dynamic that varies according to country context and gender, providing valuable insights into understanding entrepreneurial behavior in emerging economies.

Therefore, a quantitative study was conducted with a sample of 961 undergraduate and graduate business students from these countries. To analyze the data and test the hypotheses, a variance-based structural equation modeling technique (PLS-SEM) was used. Regarding the results of the proposed measurement model, all scales used in the study demonstrated validity and reliability within the expected parameters, with Cronbach’s alpha and composite reliability values above 0.7, and an average variance extracted (AVE) above 0.5. Regarding the structural results, the four proposed hypotheses were confirmed in the general sample: fear of failure negatively affected both entrepreneurial self-efficacy (β = −0.058; *p* = 0.037) and entrepreneurial passion (β = −0.112; *p* < 0.001); likewise, entrepreneurial passion showed a strong positive effect on entrepreneurial self-efficacy (β = 0.620; *p* < 0.001), which is the most significant relationship in the model, and the mediating effect of entrepreneurial passion was confirmed (β = −0.069; *p* < 0.001). However, the multi-group analysis revealed significant contextual differences: (1) Regarding gender, all relationships were significant in women, while in men, the direct effect of fear of failure on entrepreneurial self-efficacy did not reach statistical significance; (2) In country comparisons, only the passion-self-efficacy relationship was consistent across all four countries, with Mexico and Peru being where the most robust effects of the complete model were observed. The model explained 39.6% of the variance in entrepreneurial self-efficacy, providing empirical evidence of the dynamics of these psychological factors in the specific context of the studied population.

This study is based on the idea that university students in business careers, especially in emerging economies, intend to develop their businesses; therefore, they are potential entrepreneurs ([13]; [79]). Higher education institutions are aware of this, so it is more common to find academic subjects related to entrepreneurship in business careers and careers that are exclusively associated with this area. However, the various variables that drive entrepreneurial potential, such as entrepreneurial self-efficacy, are limited by the fear of failure in students ([55]; [70]), especially when the social conditions of the environment and the experiences of failure of other entrepreneurs are louder than the benefits of entrepreneurship, generating doubts about the ability to achieve success ([51]; [70]). However, the present study postulates that to the extent that an entrepreneur or potential entrepreneur generates greater passion for entrepreneurship, this fear will dissipate or be controlled, given that both are competing psychological variables. For the same reason, if the fear of failure is of greater intensity, it can affect entrepreneurial passion ([36]). Therefore, universities and higher education institutions must implement measures to foster a passion for entrepreneurship among their students. The latter is analyzed through the theories of social learning and social cognitive theories postulated by Albert Bandura, who argues that the acquisition of knowledge and understanding by people is associated with the observation or perception of what happens in the environment ([8], [9], [11]). In other words, the environment will increase the passion for entrepreneurship or, failing that, the fear of failure, which will impact the entrepreneurial self-efficacy. Therefore, the present study sought to test this postulate through a predictive model, where passion for entrepreneurship mediates the effect of fear of failure on the entrepreneurial self-efficacy of business students from four emerging economies.

The overall analysis revealed that fear of failure negatively impacts entrepreneurial self-efficacy, eroding individuals’ confidence in their entrepreneurial abilities. This finding aligns with that of [36] ([36]), who state that fear of failure decreases entrepreneurial motivation and undermines self-efficacy. Similarly, [90] ([90]) note that this fear represents a psychological barrier that limits the expectations of potential entrepreneurs in achieving their established goals. However, significant differences are observed when the results are compared between countries. The adverse effect of fear of failure on entrepreneurial self-efficacy is only significant in the overall sample. However, this relationship does not reach statistical significance when analyzed individually for each country. This variability can be explained by [14] ([14]) and [81] ([81]), who found that the environment modulates the impact of fear of failure, affecting entrepreneurial self-efficacy differently depending on the specific sociocultural context.

The results clearly show that fear of failure significantly and negatively affects entrepreneurial passion, which corroborates the findings of [6] ([6]) and [61] ([61]). These authors explain that fear of failure reduces entrepreneurial passion because it generates perceptions of a low probability of success in the face of entrepreneurial opportunities. [64] ([64]) complement this view by pointing out that when individuals experience fear, their passion and motivation for entrepreneurship decrease. This finding is especially evident in Mexico and Peru, where the adverse effects are statistically significant. However, the data show apparent differences by gender: women exhibit greater resistance to the negative impact of fear of failure on entrepreneurial passion, suggesting that they have developed more effective coping mechanisms than men. This can be interpreted from the perspective of [91] ([91]), who emphasized the importance of creating resilient enabling environments to preserve entrepreneurial passion.

The relationship between entrepreneurial passion and entrepreneurial self-efficacy is consistent across all countries studied. Entrepreneurial passion significantly strengthens entrepreneurial self-efficacy, consistent with [81] ([81]), who state that the higher the entrepreneurial passion, the higher the entrepreneurial self-efficacy. [98] ([98]) complement this idea by pointing out that entrepreneurial passion is an element that strengthens entrepreneurial self-efficacy when facing challenges. This finding supports the premise that enthusiasm and commitment to entrepreneurial activities reinforce confidence in one’s capabilities for entrepreneurship, confirming [8]’s ([8], [9], [11]) social cognitive theory, which posits that positive emotional experiences strengthen one’s self-efficacy beliefs.

Similarly, the findings reinforce [97] ([97]), who found that entrepreneurial passion plays a vital role in strengthening entrepreneurial self-efficacy and increasing the probability of success in entrepreneurs. [25] ([25]) explain this phenomenon by noting that an entrepreneur’s credibility in themselves generates a special dynamic that arouses passion, expressed in a commitment to face challenges. This positive relationship represents the strongest link in our model, with a standardized coefficient of 0.620, highlighting the importance of fostering entrepreneurial passion in educational programs.

Mediation analysis reveals that entrepreneurial passion serves as a bridge between fear of failure and entrepreneurial self-efficacy. Fear of failure has a significant indirect effect on entrepreneurial self-efficacy through its influence on entrepreneurial passion. This can be explained from the perspective of [17] ([17]) and [3] ([3]), who state that passion can mitigate the barriers imposed by fear, thus generating a positive mindset. This finding is consistent with the dual model of passion described by [36] ([36]) and [48] ([48]), which states that harmonious passion is associated with favorable outcomes such as persistence and success in entrepreneurial initiatives. However, the mediating role of entrepreneurial passion varies by country. In Mexico and Peru, the mediating effect is significant, while in Colombia and Ecuador, it does not reach statistical significance. This indicates that contextual factors, such as economic, cultural, and educational conditions, influence the interaction between entrepreneurial passion, fear of failure, and entrepreneurial self-efficacy. As [50] ([50]) point out, the behavior of entrepreneurs in adverse situations may vary depending on the context, which explains the differences found among the countries studied.

Regarding gender differences, mediation is present in both women and men, but with some particularities. In women, all the relationships in the model are significant, while in men, the direct effect of fear of failure on entrepreneurial self-efficacy does not reach statistical significance. This suggests that women are more sensitive to the direct impact of fear of failure on their self-efficacy. However, they may benefit more from the protective effects of entrepreneurial passion. This can be interpreted from the perspective of [44] ([44]), who explained that university students tend to develop greater passion for entrepreneurial activities and strengthen their skills upon receiving entrepreneurial education.

Cross-country comparisons reveal that in Colombia and Ecuador, fear of failure does not significantly affect entrepreneurial self-efficacy or passion. This can be interpreted in light of the theory of protective motivation, as mentioned by [47] ([47]), which posits that a fear of failure in specific contexts can motivate individuals to adopt protective behaviors in response to perceived threats. This dynamic could operate differently in these countries, generating adaptive responses to the fear of failure.

En México, el efecto negativo del miedo al fracaso sobre la pasión emprendedora es más fuerte que en otros países. Esto podría explicarse por factores específicos de ese país, como señala ([92]), quienes explican que cuando las condiciones son inciertas, el miedo al fracaso emerge, inhibiendo la iniciativa emprendedora. Sin embargo, en México también se observa un fuerte efecto mediador de la pasión emprendedora, lo que sugiere que, a pesar del mayor impacto del miedo, la pasión juega un papel crucial como mecanismo protector.

In Mexico, the adverse effect of fear of failure on entrepreneurial passion is more substantial than in other countries. This could be explained by factors specific to that country, as pointed out by [92] ([92]), who explain that when conditions are uncertain, fear of failure emerges, inhibiting entrepreneurial initiative. However, in Mexico, a strong mediating effect of entrepreneurial passion is also observed, suggesting that, despite the greater impact of fear, passion plays a crucial role as a protective mechanism.

Finally, the consistency of the positive effect of entrepreneurial passion on entrepreneurial self-efficacy across all countries studied reinforces the theory of planned behavior mentioned by [103] ([103]) and [68] ([68]), who state that attitudes, subjective norms, and perceived behavioral control influence individuals’ intentions. In this sense, entrepreneurial passion acts as a catalyst that enhances perceived control and self-efficacy, thus facilitating the transition from entrepreneurial intention to entrepreneurial action.

### Theoretical and Managerial Implications

From a theoretical perspective, this study contributes new insights to the literature on entrepreneurship by demonstrating that the relationship between fear of failure and self-efficacy varies across different contexts (countries), thereby contradicting the notion of a universal effect. The results confirmed the dual model of passion described by [36] ([36]) and [48] ([48]), showing that harmonious passion is associated with favorable outcomes such as persistence and success in entrepreneurial initiatives. Likewise, the research reinforces the theory of entrepreneurial self-efficacy proposed by [12] ([12]) and [76] ([76]), confirming that it is an effective means of moving the entrepreneur’s knowledge into concrete action. The study also validates the relevance of the socio-cognitive theory in the specific context of emerging economies, where learning arises due to the dynamic interaction between personal and environmental factors, as suggested by [82] ([82]). Additionally, this research provides significant empirical evidence of gender differences in the entrepreneurial process, expanding knowledge on how men and women respond differently to the fear of failure in entrepreneurial contexts.

At the practical level, the findings indicate that educational institutions should develop programs specifically designed to strengthen entrepreneurial passion as a mechanism to effectively counteract the fear of failure, with particular emphasis on countries such as Mexico and Peru, where the negative relationship between fear of failure and entrepreneurial passion is more evident. Entrepreneurial training must incorporate gender-differentiated components, leveraging the greater resistance to fear of failure observed in women. It is essential to create learning environments that simulate real entrepreneurial scenarios, as proposed by [53] ([53]), to correct attribution to failure and significantly increase self-efficacy in authentic situations. Likewise, universities must adopt the responsibility of implementing comprehensive programs that strengthen technical knowledge and the psychological aspects of entrepreneurship, as suggested by [13] ([13]) and [79] ([79]) in their research. Additionally, entrepreneurship programs should systematically expose students to successful business models, as this strategy significantly improves confidence, as documented by [16] ([16]) and [63] ([63]) in their studies on entrepreneurship education.

From a managerial perspective, organizations must develop effective mechanisms to identify and cultivate entrepreneurial passion within their teams and create environments that normalize failure as a natural part of the learning process. It is highly recommended that mentoring structures be implemented, where experienced entrepreneurs can guide and help develop self-efficacy in new entrepreneurs, thereby transmitting both tacit and explicit knowledge. Business incubators and accelerators should incorporate components that deliberately strengthen entrepreneurial passion and self-efficacy, thereby transcending an exclusive focus on technical aspects of the business. In this sense, companies should seriously consider harnessing the entrepreneurial potential of women, given their greater resistance to the fear of failure, as demonstrated in this study. Ultimately, organizational leaders are responsible for fostering corporate cultures that actively encourage experimentation and normalize failure as an integral part of business learning, particularly in emerging economies, where risk aversion can be a significant barrier to innovation and entrepreneurship.

## 6. Conclusions

This study examines the interactions between fear of failure, entrepreneurial self-efficacy, and entrepreneurial passion in university business students from four emerging economies in Latin America. The results provide valuable insights into the psychological dynamics underlying entrepreneurial behavior in these contexts, with significant implications for entrepreneurship education and policy promotion.

The data demonstrate that fear of failure negatively affects entrepreneurial self-efficacy in the overall sample but with significant variations by country and gender. This adverse effect is only substantial when analyzing the entire sample, while it does not reach statistical significance in the country-by-country analyses. This variability reflects how country-specific contextual factors influence the way fear of failure affects perceptions of entrepreneurial capability.

The research reveals that fear of failure hinders entrepreneurial passion, particularly in Mexico and Peru. This reflects how negative emotions associated with fear can diminish enthusiasm and commitment to entrepreneurship. However, it notes that there are gender differences, with women showing greater resistance to this effect, which points to possible distinct coping mechanisms that warrant further investigation.

Another finding is the strong positive relationship between entrepreneurial passion and entrepreneurial self-efficacy, which is consistent across all countries and genders studied. This demonstrates that when students develop enthusiasm and commitment to entrepreneurial activities, their confidence in their entrepreneurial abilities is significantly strengthened. The consistency of this relationship across different cultural contexts reveals a universal pattern that can be leveraged in educational programs.

This study confirms the mediating role of entrepreneurial passion in the relationship between fear of failure and entrepreneurial self-efficacy. This mediating mechanism reveals that although fear of failure can negatively affect entrepreneurial self-efficacy, this effect can be mitigated by developing strong entrepreneurial passion. This finding provides evidence for educational interventions that strengthen entrepreneurial passion as a protective factor against fear of failure.

The differences between countries demonstrate that cultural, economic, and educational contexts are vital for shaping entrepreneurial attitudes. The lower sensitivity to fear of failure in Colombia and Ecuador suggests the presence of specific protective factors in these environments, which warrant further investigation.

The results also reveal gender differences. Women exhibit a distinct dynamic compared to men in the relationship between fear of failure, entrepreneurial passion, and self-efficacy. This demonstrates that women can develop more effective coping mechanisms in the face of uncertainty and the associated risks of entrepreneurship. This represents a potential advantage that should be considered in policies aimed at promoting female entrepreneurship.

### 6.1. Limitations

The main limitation of this study is the generalizability of the results, which is limited by the nature of the sample, focusing exclusively on business students in Latin American universities. This may not adequately represent the cultural and contextual diversity of other regions and entrepreneurs outside the academic field.

The cross-sectional design limits our ability to establish causal relationships between the variables studied. Fear of failure, self-efficacy, and entrepreneurial passion may influence each other in various ways over time, aspects that can only be captured through a longitudinal approach.

Another limitation is the potential response bias inherent in self-reported surveys. Although validated scales with appropriate translation procedures were used, participants may have responded in a socially desirable manner, affecting the validity of the results, especially for sensitive constructs, such as fear of failure. Finally, although the structural model demonstrated general robustness, the variability of the results between countries suggests the need to incorporate additional variables that better capture the contextual factors specific to each national environment, such as indicators of economic stability or the development of the entrepreneurial ecosystem.

### 6.2. Directions for Future Research

For future research, it is recommended that the geographic scope be expanded to include participants from diverse regions and cultures, which would enable a more comprehensive understanding of how the studied factors interact in various socioeconomic contexts. This is particularly relevant considering the significant differences between the four countries in this study.

A longitudinal design could provide insights into how the relationships between fear of failure, self-efficacy, and entrepreneurial passion evolve throughout the entrepreneurial process, from the initial intention to business consolidation. This approach would facilitate a deeper understanding of the dynamic nature of these psychological variables.

Finally, studies should be conducted on specific interventions designed to strengthen entrepreneurial passion and evaluate their effectiveness in mitigating fear of failure and enhancing entrepreneurial self-efficacy. This could include specific educational programs, mentoring, and psychological support tailored to different demographic groups.

## Figures and Tables

**Figure 1 behavsci-15-00951-f001:**
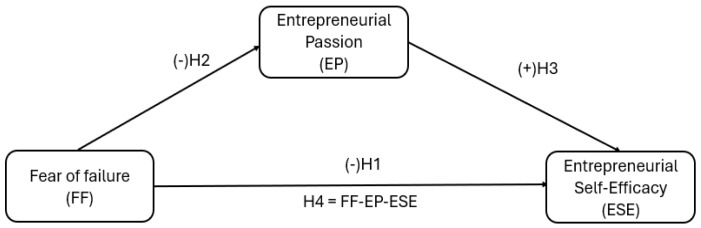
Integrated conceptual model.

**Figure 2 behavsci-15-00951-f002:**
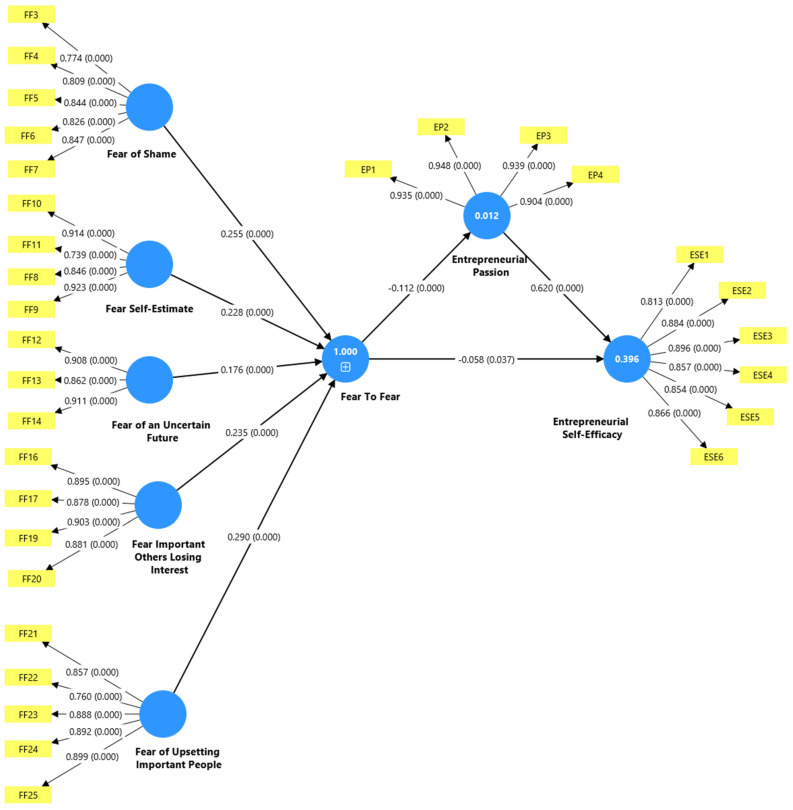
The structural model for the general sample.

**Table 1 behavsci-15-00951-t001:** Socio-demographic data (n = 961).

Variable	Categories	Frequency	%
Gender	Male	398	41.4
	Female	563	58.6
Country	Colombia	219	22.8
	Ecuador	230	23.9
	Mexico	236	24.6
	Peru	276	28.7
Age	Under 21 years old	301	31.3
	21 to 25 years old	428	44.5
	26 to 30 years old	73	7.6
	31 to 35 years old	55	5.7
	36 to 40 years old	37	3.9
	More than 40 years old	67	7.0
Academic Level	Undergraduate	845	87.9
	Master’s degree	103	10.7
	Doctorate	13	1.4

**Table 2 behavsci-15-00951-t002:** Scales’ validity and reliability.

Variables	Code	Mean	SD	SK	K	Loading	α	CR	AVE
Entrepreneurial Passion (EP)	EP1	5.95	1.47	−1.55	1.79	0.934	0.949	0.963	0.868
	EP2	6.13	1.41	−1.87	2.97	0.948			
	EP3	6.1	1.36	−1.78	2.67	0.939			
	EP4	5.77	1.52	−1.33	1.10	0.905			
Entrepreneurial Self-Efficacy (ESE)	ESE1	5.09	1.52	−0.64	−0.11	0.813	0.931	0.946	0.743
	ESE2	4.86	1.63	−0.47	−0.55	0.885			
	ESE3	5.04	1.56	−0.61	−0.32	0.896			
	ESE4	4.99	1.59	−0.61	−0.32	0.856			
	ESE5	4.95	1.58	−0.58	−0.41	0.853			
	ESE6	5.21	1.44	−0.63	−0.07	0.867			
Fear shame (FS)	FF3	2.42	1.28	0.46	−0.89	0.774	0.878	0.911	0.673
	FF4	2.46	1.24	0.49	−0.69	0.809			
	FF5	2.39	1.3	0.55	−0.83	0.844			
	FF6	2.26	1.27	0.63	−0.77	0.826			
	FF7	2.42	1.39	0.51	−1.04	0.847			
Fear self-estimate (FSE)	FF8	1.92	1.2	1.15	0.25	0.846	0.878	0.918	0.737
	FF9	2.01	1.22	1.01	−0.05	0.923			
	FF10	2.05	1.23	0.95	−0.19	0.914			
	FF11	2.61	1.38	0.36	−1.11	0.739			
Fear of an Uncertain Future (FUF)	FF12	2.39	1.29	0.57	−0.76	0.908	0.874	0.923	0.799
	FF13	2.65	1.31	0.29	−1.03	0.862			
	FF14	2.39	1.27	0.57	−0.72	0.911			
Fear Important Others Losing Interest (FIOLI)	FF16	2.3	1.31	0.64	−0.75	0.895	0.912	0.938	0.791
	FF17	2.37	1.3	0.56	−0.83	0.878			
	FF19	2.36	1.25	0.59	−0.64	0.903			
	FF20	2.3	1.27	0.61	−0.75	0.881			
Fear of upsetting important people (FUIO)	FF21	2.18	1.24	0.78	−0.47	0.857	0.911	0.934	0.741
	FF22	2.42	1.31	0.52	−0.88	0.760			
	FF23	2.09	1.24	0.94	−0.17	0.888			
	FF24	2.23	1.23	0.71	−0.5	0.892			
	FF25	2.2	1.26	0.79	−0.45	0.899			

Note: SD = Standard Deviation; α = Cronbach’s Alpha; CR = Composite Reliability; AVE = Average Variance Extracted; SK = Skewness; K = Kurtosis.

**Table 3 behavsci-15-00951-t003:** Discriminant validity (Fornell-Larcker criteria).

	1	2	3	4	5	6	7	8
Entrepreneurial Passion (EP)	0.932							
Entrepreneurial Self-Efficacy (ESE)	0.627	0.862						
Fear of Failure (FF)	−0.112	−0.127	0.727					
FIOLI	−0.043	−0.081	0.835	0.890				
FSE	−0.147	−0.143	0.850	0.582	0.858			
FS	−0.077	−0.107	0.841	0.593	0.662	0.820		
FUF	−0.112	−0.174	0.802	0.544	0.744	0.642	0.894	
FUIO	−0.099	−0.060	0.875	0.759	0.638	0.631	0.569	0.861

**Table 4 behavsci-15-00951-t004:** Heterotrait-monotrait ratio (HTMT).

	Original Sample (O)	Sample Mean (M)	2.5%	97.5%
Entrepreneurial Self-Efficacy (ESE) → Entrepreneurial Passion (EP)	0.662	0.662	0.611	0.709
Fear of Failure (FF) → Entrepreneurial Passion (EP)	0.116	0.121	0.073	0.181
Fear of Failure (FF) → Entrepreneurial Self-Efficacy (ESE)	0.133	0.138	0.085	0.205
FIOLI → EP	0.049	0.059	0.028	0.116
FIOLI → ESE	0.085	0.089	0.037	0.159
FSE → EP	0.159	0.160	0.094	0.228
FSE → ESE	0.157	0.161	0.100	0.232
FSE → FIOLI	0.650	0.650	0.595	0.707
FS → EP	0.084	0.089	0.040	0.152
FS → ESE	0.115	0.119	0.061	0.189
FS → FIOLI	0.661	0.661	0.605	0.714
FS → FSE	0.756	0.756	0.708	0.801
FUF → EP	0.119	0.122	0.065	0.184
FUF → ESE	0.188	0.189	0.118	0.260
FUF → FIOLI	0.606	0.607	0.544	0.664
FUF → FSE	0.849	0.849	0.810	0.885
FUIO → EP	0.105	0.109	0.057	0.173
FUIO → ESE	0.065	0.075	0.041	0.134
FUIO → FIOLI	0.831	0.831	0.789	0.869
FUIO → FSE	0.712	0.712	0.657	0.763
FUIO → FS	0.705	0.704	0.654	0.751
FUIO → FUF	0.635	0.635	0.576	0.688

**Table 5 behavsci-15-00951-t005:** PLS path model main effects.

General Sample	Beta	SD	t	*p*
FIOLI → FF	0.235	0.005	47.824	0.000
FSE → FF	0.228	0.005	47.378	0.000
FS → FF	0.255	0.006	43.076	0.000
FUF → FF	0.176	0.005	38.09	0.000
FUIO → FF	0.29	0.006	50.26	0.000
H1: FF → ESE	−0.058	0.028	2.087	0.037
H2: FF → EP	−0.112	0.031	3.597	0.000
H3: EP → ESE	0.620	0.025	25.178	0.000
H4: FF→ EP→ ESE	−0.069	0.019	3.579	0.000

Standard Deviation (SD), *p*-values (*p*), T statistics (t).

**Table 6 behavsci-15-00951-t006:** PLS path model for gender.

Men	b	SD	t	*p*
FIOLI → FF	0.239	0.008	29,811	0.000
FSE → FF	0.221	0.008	28,685	0.000
FS → FF	0.25	0.008	29,477	0.000
FUF → FF	0.17	0.006	26,595	0.000
FUIO → FF	0.289	0.009	31,075	0.000
H1: FF → ESE	−0.051	0.046	1105	0.269
H2: FF → EP	−0.160	0.048	3355	0.001
H3: EP→ ESE	0.531	0.045	11,698	0.000
H4: FF → EP→ ESE	−0.085	0.026	3221	0.001
**Women**	**b**	**SD**	**t**	** *p* **
FIOLI → FF	0.232	0.006	36,731	0.000
FSE → FF	0.234	0.006	36,97	0.000
FS → FF	0.258	0.008	32,648	0.000
FUF → FF	0.18	0.006	27,926	0.000
FUIO → FF	0.29	0.007	39,626	0.000
H1: FF → ESE	−0.129	0.047	2755	0.006
H2: FF → EP	−0.092	0.042	2192	0.028
H3: EP → ESE	0.667	0.028	24,019	0.000
H4: FF → EP → ESE	−0.061	0.028	2205	0.027

**Table 7 behavsci-15-00951-t007:** PLS path model for each country.

Colombia	β	SD	t	*p*
FIOLI → FF	0.236	0.011	22.172	0.000
FSE → FF	0.219	0.009	24.858	0.000
FS → FF	0.266	0.012	22.560	0.000
FUF → FF	0.182	0.009	21.055	0.000
FUIO → FF	0.291	0.013	22.631	0.000
H1: FF → ESE	−0.079	0.043	1855	0.064
H2: FF → EP	0.036	0.068	0.532	0.595
H3: EP → ESE	0.740	0.035	21,409	0.000
H4: FF → EP → ESE	0.027	0.050	0.531	0.596
**Ecuador**	**b**	**SD**	**t**	** *p* **
FIOLI → FF	0.234	0.009	25.084	0.000
FSE → FF	0.233	0.010	23.276	0.000
FS → FF	0.256	0.012	22.108	0.000
FUF → FF	0.173	0.012	14.736	0.000
FUIO → FF	0.285	0.011	25.469	0.000
H1: FF → ESE	−0.043	0.068	0.639	0.523
H2: FF → EP	−0.127	0.068	1861	0.063
H3: EP → ESE	0.450	0.056	8007	0.000
H4: FF → EP → ESE	−0.057	0.032	1793	0.073
**Mexico**	**b**	**SD**	**t**	** *p* **
FIOLI → FF	0.231	0.013	18.437	0.000
FSE → FF	0.244	0.013	18.387	0.000
FS → FF	0.245	0.016	15.455	0.000
FUF → FF	0.188	0.011	17.220	0.000
FUIO → FF	0.304	0.014	21.201	0.000
H1: FF → ESE	−0.086	0.060	1421	0.155
H2: FF → EP	−0.260	0.072	3626	0.000
H3: EP → ESE	0.570	0.056	10,149	0.000
H4: FF → EP → ESE	−0.148	0.047	3132	0.002
**Perú**	**b**	**SD**	**t**	** *p* **
FIOLI → FF	0.235	0.010	22.795	0.000
FSE → FF	0.220	0.008	28.183	0.000
FS → FF	0.262	0.011	24.688	0.000
FUF → FF	0.160	0.008	20.366	0.000
FUIO → FF	0.288	0.010	27.568	0.000
H1: FF → ESE	−0.054	0.065	0.840	0.401
H2: FF → EP	−0.202	0.052	3886	0.000
H3: EP → ESE	0.437	0.071	6142	0.000
H4: FF → EP → ESE	−0.088	0.023	3794	0.000

## Data Availability

The original contributions presented in this study are included in this article. Further inquiries should be directed to the corresponding author.

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
