# Peer review of "The Interaction Between Self-Efficacy, Fear of Failure, and Entrepreneurial Passion: Evidence from Business Students in Emerging Economies"

_behavsci, 2025, doi:10.3390/bs15070951_

Round 1

Reviewer 1 Report

Comments and Suggestions for Authors

The research problem is well articulated and well developed. The authors should double-check some of the in-text citations, such as C. Duong & Vu, 2023a, J. Schmutzler,  and correct the in-text initials. Despite the slight conceptual overlap; the constructs have been well delineated and discussed. This is a good contribution to knowledge, although the authors have a theoretical contribution section, there is no mention of actual theories, this should be corrected. There are a couple of repeated references (a) (b) of the exact same paper, this needs to be corrected across the paper:

Sahid, S., Norhisham, N., & Narmaditya, B. (2024a). Interconnectedness between entrepreneurial self-efficacy, attitude, and business creation: A serial mediation of entrepreneurial intention and environmental factor. Heliyon, 10(9). https://doi.org/10.1016/j.heliyon.2024.e30478 958

Sahid, S., Norhisham, N., & Narmaditya, B. (2024b). Interconnectedness between entrepreneurial self-efficacy, attitude, and business creation: A serial mediation of entrepreneurial intention and environmental factor. Heliyon, 10(9). https://doi.org/10.1016/j.heliyon.2024.e30478

Schmutzler, J., Andonova, V., & Diaz-Serrano, L. (2019a). How Context Shapes Entrepreneurial Self-Efficacy as a Driver of Entrepreneurial Intentions: A Multilevel Approach. Entrepreneurship: Theory and Practice, 43(5), 880–920. https://doi.org/10.1177/1042258717753142

Schmutzler, J., Andonova, V., & Diaz-Serrano, L. (2019b). How Context Shapes Entrepreneurial Self-Efficacy as a Driver of Entrepreneurial Intentions: A Multilevel Approach. Entrepreneurship Theory and Practice, 43(5), 880–920. https://doi.org/10.1177/1042258717753142

Schmutzler, J., Andonova, V., & Diaz-Serrano, L. (2019c). How Context Shapes Entrepreneurial Self-Efficacy as a Driver of Entrepreneurial Intentions: A Multilevel Approach. Entrepreneurship Theory and Practice, 43(5), 880–920. https://doi.org/10.1177/1042258717753142

These are just examples but there is more. 

Author Response

Dear Reviewer,

We extend our sincere gratitude for your insightful comments, which have been invaluable in enhancing the quality of our manuscript. Your thoughtful feedback has contributed significantly to refining our work, and we have made concerted efforts to address each of your suggestions.

We are optimistic that this revised version of the paper now meets the anticipated standards for publication in this esteemed journal. Below is a comprehensive list of responses addressing your comments and suggestions.

Thank you once again for your time and expertise.

Best regards,

Reviewer 2 Report

Comments and Suggestions for Authors

Dear Authors,

Thank you for providing the opportunity to review your submission, entitled The Role of Self-Efficacy in the Relationship Between Fear of Failure and Entrepreneurial Passion: Evidence from Students in Emerging Economies. While certain aspects of the paper hold merit, I believe that the manuscript requires significant revisions to address the following substantial concerns.

Abstract. The stated aim of the paper, "is to construct a predictive model to examine the influence of fear of failure and entrepreneurial passion on the entrepreneurial self-efficacy of university students" (page 1, lines 20-21), appears to contradict the title, which refers to the role of Self-Efficacy in the Relationship Between Fear of Failure and Entrepreneurial Passion. Figure 1, titled Integrated Conceptual Model, aligns, in my opinion, with the title but not with the stated aim in the Abstract. The specification that the research concerns business students should be included in the title and consistently addressed throughout the paper, given the theme of the study, which pertains to entrepreneurship. The statement on lines 24-25, "Scales to measure fear of failure, entrepreneurial self-efficacy and entrepreneurial passion were translated into Spanish and validated," lacks support in the paper. The authors seem to conflate back-translation with the validation of a scale, which is a very complex process. Furthermore, the paper does not present even a single item from any of the mentioned scales. Not all major conclusions, such as those regarding gender (cf. Chapter 6: Conclusions), are presented clearly and objectively. Additionally, the statement "this study underscores the need to design educational (lines 32-33)" is not substantiated in Chapter 6: Conclusions.

Introduction. In general, this section does not provide an adequate context and does not clearly establish the direction of the research. Among other things, a brief review of previous studies concerning research on the entrepreneurial antecedents, such as fear of failure, entrepreneurial passion on the entrepreneurial self-efficacy among business students relevant in the literature is missing. Moreover, it could probably have been observed that most of these studies largely focus on entrepreneurial intention as a predictor of entrepreneurial behavior. In the absence of the dependent variable "entrepreneurial intention," in my opinion, it is very difficult to discuss, for instance, entrepreneurial passion among students. I recommend that in the revised paper, the authors clearly and concisely state the main purpose of the study, the specific questions they aim to answer, the fundamental hypotheses, and briefly describe the theoretical and practical contributions brought by the paper.

Theoretical background. A general theoretical framework is missing, but it is necessary for scientific papers.  

Method. I believe that a title such as Research Methodology or Study Methodology would be more relevant, especially since this chapter, in my view, has certain vulnerabilities.

Specifically, in subsection 3.2, Instrument, the authors do not provide the necessary details for a scientific paper, such as the source of the scales, the number of items, examples of items, details regarding the compilation of items into scales, etc. The statement that the scales were revalidated (p. 8, l. 33) is entirely unsupported. Only a detailed presentation of the questionnaire and a clear distinction between items, scales, and variables could facilitate comparisons with other studies and the generalization of the results. Table 2 on page 9 does not indicate its source, and columns 3 and 4 create confusion, starting with their titles. For instance, in columns 3 and 4, what variables are being referred to? Previously, you described three variables extensively, starting on page 3, line 108, subsection 2.1. Research Variables. I recommend revising and correcting the abbreviation in the table for Entrepreneurial Passion—it should likely be EP instead of PA.

So, I suggest providing greater detail on the following: a. the operationalization of the three investigated variables, preferably in the form of a table with at least three columns, namely: the investigated variables/dimensions, items (at least their number, if they cannot be fully listed), and authors; b. data preparation and descriptive statistics.

  1. Results; Discussions. Given that this part of the paper is currently difficult to follow (pp. 9-19), I suggest rewriting to allow readers to more easily follow the flow of information. In the Results section, the author should focus solely on reporting the data (with accuracy and clarity), while in the Discussion section, the interpretations (both objective and subjective) of the results should be presented. This approach would provide a more comprehensive view and place the results in a broader context, especially through comparisons both between the four countries, and with other studies from different countries, highlighting the possibility of generating additional findings.

The authors should reconsider the theoretical, practical, and managerial implications (pp. 17-19) and provide more concrete results in light of the revised paper.

  1. Conclusions. The Conclusion chapter of a scientific article should provide a clear and concise ending to the research, highlighting a summary of the main findings. Now, the input of the paper is unclear. The paper needs to be rewritten to address behavioral sciences.

Additionally, I strongly recommend paying attention to the following aspects:

  1. Conceptual issues: What is the theoretical foundation for using fear of failure, entrepreneurial passion, and entrepreneurial self-efficacy for business students across the three education cycles: bachelor's, master's, and doctoral studies? Do doctoral students have the same entrepreneurial exposure as those in the first cycle—undergraduate university studies? You must consider all this and much more when making claims, such as "solid conceptual framework," as stated on page 18, lines 617-618. Start your presentation of each of the three variables with a precise definition of the concepts and provide the authors, the work, and the page where the definitions can be found. Also, provide a solid justification for using only independent variables, as the study does not consider any dependent variable, such as business students` entrepreneurial intention.
  2. Methodological issues: Outline the development of the questionnaire. Specify the types of questions (e.g., Likert scale, closed-ended, demographic) and include a rationale for each question type. Highlight whether the questionnaire was pilot-tested for reliability and validity. Explain how each variable was measured or operationalized, and discuss the steps taken to minimize potential biases during the study.
  3. References: Indicate recent literature relevant to the topic of the paper and provide a critical analysis of it. Please correct all errors related to the bibliography. Some references are not included in the References section, e.g., Malhotra (2008), p.7, r. 307. Other references have been misunderstood. In the References section, starting on page 22, many titles are repeated. For example, the same work appears twice:
  • Anjum, T., Ramzani, S., Farrukh, M., Raju, V., Nazar, N., & Shahzad, I. (2018a). Entrepreneurial Intentions of Pakistani Students: The Role of Entrepreneurial Education, Creativity Disposition, Invention Passion & Passion for Founding. Journal of Management Research, 10(3), 76. https://doi.org/10.5296/jmr.v10i3.13253
  • Anjum, T., Ramzani, S., Farrukh, M., Raju, V., Nazar, N., & Shahzad, I. (2018b). Entrepreneurial Intentions of Pakistani Students: The Role of Entrepreneurial Education, Creativity Disposition, Invention Passion & Passion for Founding. Journal of Management Research, 10(3), 76. https://doi.org/10.5296/jmr.v10i3.13253.

It is well-known, for instance, that "2018a" refers to the author's first work published in 2018, while "2018b" refers to the author's second work published in the same year, 2018.

  1. Author Contributions: Review the author contributions and ensure, from all perspectives, that the author(s) responsible for rewriting and supervision have implemented a coherent and unified approach across all sections of the paper, including the preparation of tables. I strongly reccomand to keep only the most relevant tables, transferring necessary data that would overburden the paper into the Appendix, as this would make it difficult for readers to read and understand. In its current form, the tables lack sources and proper explanations.

I wish you success in your re-written process.

Comments on the Quality of English Language

The quality and use of the English language need significant improvement.

Author Response

(The authors gave the same response as above.)

Reviewer 3 Report

Comments and Suggestions for Authors

Please find attached all my comments and suggestions related to your manuscript

Author Response

(The authors gave the same response as above.)

Round 2

Reviewer 2 Report

Comments and Suggestions for Authors

Dear authors, congratulations on the revised version of the article. I recommend its publication after another review, especially in terms of its translation/writing in English. I say this also because, on page 6, lines 286-289, the Spanish version remains.

Author Response

Dear Reviewer,

We express our sincere gratitude for your insightful comments. 
We have sent the English for review to a specialist, and you can see the improvement in the whole paper.

We hope this revised paper version meets the expected requirements for publication in this prestigious journal.

Reviewer 3 Report

Comments and Suggestions for Authors

The authors have done a very good job in addressing the comments and suggestions. This version has improved significantly, which only requires some revisions as follows:

I suggest including the effect size to check the inner model.

There are some formatting issues with in-text citations. For example, in Line 99: “According to (Eisenmann, 2021)” should follow proper citation format “According to Eisenmann (2021)”. Please ensure that all in-text citations follow a coherent structure.

Please rephrase the title of 5.1. subsection as “Theoretical, and Managerial Implications”.

Please correct the title in line 515 “Hypothesis Development”, by using “Hypotheses testing results”.

Some English editing is needed to correct the various grammatical errors and typos present in the manuscript (lines 505, 515, 574, 549, etc). I strongly recommend a professional proofreading for the entire manuscript.

Wish you all the best in your research

Author Response

Dear Reviewer,
We are very thankful for your insightful comments. 
We have sent a specialist to review the English.
The following explains how we have improved the document:

Comment 1: I suggest including the effect size to check the internal model.

Response 1: We have included the explanation of the effect size. You can see this improvement on lines 549 to 557.

Comment 2 : There are some formatting issues with in-text citations. For example, in Line 99: “According to (Eisenmann, 2021)” should follow proper citation format “According to Eisenmann (2021)”. Please ensure that all in-text citations follow a coherent structure.

Response 2: We have revised the citation format throughout the document; you can see the improvement throughout the document.

Comment 3: Please rephrase the title of 5.1. subsection as “Theoretical, and Managerial Implications”.

Response 3: We have improved this subtitle; you can see it on line 740.

Comment 4 : Please correct the title in line 515 “Hypothesis Development”, by using “Hypotheses testing results”.

Response 4: We have improved that title, as shown on line 511.

Comment 5: Some English editing is needed to correct the various grammatical errors and typos present in the manuscript (lines 505, 515, 574, 549, etc). I strongly recommend a professional proofreading for the entire manuscript.

Response 5: The paper has been reviewed by a specialist; you can see the improvement throughout.

Comment 6: Wish you all the best in your research

Response 6: Thank you very much for your comments. We hope this revised paper version meets the expected requirements for publication in this prestigious journal.